# Thermal Conductivity of Aluminum Alloys—A Review

**DOI:** 10.3390/ma16082972

**Published:** 2023-04-08

**Authors:** Ailing Zhang, Yanxiang Li

**Affiliations:** 1School of Materials Science and Engineering, Tsinghua University, Beijing 100084, China; zal19@mails.tsinghua.edu.cn; 2Key Laboratory for Advanced Materials Processing Technology, Ministry of Education, Beijing 100084, China

**Keywords:** aluminum alloys, thermal conductivity, alloying elements, secondary phases, temperature, processes

## Abstract

Aluminum alloys have been extensively used as heatproof and heat-dissipation components in automotive and communication industries, and the demand for aluminum alloys with higher thermal conductivity is increasing. Therefore, this review focuses on the thermal conductivity of aluminum alloys. First, we formulate the theory of thermal conduction of metals and effective medium theory, and then analyze the effect of alloying elements, secondary phases, and temperature on the thermal conductivity of aluminum alloys. Alloying elements are the most crucial factor, whose species, existing states, and mutual interactions significantly affect the thermal conductivity of aluminum. Alloying elements in a solid solution weaken the thermal conductivity of aluminum more dramatically than those in the precipitated state. The characteristics and morphology of secondary phases also affect thermal conductivity. Temperature also affects thermal conductivity by influencing the thermal conduction of electrons and phonons in aluminum alloys. Furthermore, recent studies on the effects of casting, heat treatment, and AM processes on the thermal conductivity of aluminum alloys are summarized, in which processes mainly affect thermal conductivity by varying existing states of alloying elements and the morphology of secondary phases. These analyses and summaries will further promote the industrial design and development of aluminum alloys with high thermal conductivity.

## 1. Introduction

Aluminum has a thermal conductivity of 237 W m^−1^ K^−1^. Its density is 2.7 g cm^−3^, about one-third of the densities of steels and cast irons [1,2,3]. Aluminum alloys usually contain Si, Cu, Mg, Zn, and other alloying elements. They have the advantages of being lightweight, having good electrical and thermal conductivity, and having excellent mechanical properties [4,5]. Concerning the increasing crisis of global warming, most automotive companies manufacture auto applications using lightweight aluminum alloys to reduce the weight of automobiles, thereby reducing energy consumption and pollution [3,6,7]. In communication, base station radiators are commonly made of lightweight die casting aluminum alloys [8]. The thermal conductivity of aluminum alloys is an important performance parameter in these fields. Automotive heatproof components, such as engine blocks and cylinder heads, must possess high thermal conductivity to transfer heat quickly and uniformly to ensure the regular operation of automobiles [9]. With the development of communication systems from 4G to 5G, the heat generated in base stations increases dramatically, and the temperature of chips rises rapidly. If the heat does not dissipate quickly, it will reduce the performance and service life of base stations [10]. Based on current demands, it is significant to study the thermal conductivity of aluminum alloys to fabricate high thermal conductivity aluminum alloys that fit mechanical property requirements simultaneously.

Based on alloying elements, cast aluminum alloys can be divided into six main series: Al-Si, Al-Cu, Al-Cu-Si, Al-Mg, Al-Zn-Mg, and Al-Sn alloys [2]. Cast Al-Si alloys are the most widely used aluminum alloys, whose production output accounts for 80~90% of the world’s aluminum castings [11]. Based on the Si content, cast Al-Si alloys can be classified as hypoeutectic, eutectic, and hypereutectic Al-Si alloys [4]. Table 1 presents the composition, thermal conductivity, and tensile strength of typical cast Al-Si alloys at 298 K [2]. Automotive engine blocks and cylinder heads are commonly made of 319 and 380 alloys with excellent mechanical properties, and they have a low thermal conductivity of 109 and 96 W m^−1^ K^−1^, respectively. Table 1 shows that higher tensile strength corresponds to lower thermal conductivity in Al-Si alloys. This is because all of the strengthening methods of aluminum alloys impair thermal conductivity to some extent, including work hardening, solid solution strengthening, precipitation and dispersion strengthening, and second-phase strengthening [12,13,14,15,16,17,18,19,20,21]. Therefore, aluminum alloys should be fabricated with strengthening methods that minimize the detrimental effect on thermal conductivity to satisfy industrial demands for thermal conductivity and mechanical properties.

The addition of alloying elements to aluminum can enhance the strength of pure aluminum, followed by reducing thermal conductivity [4]. Alloying elements in aluminum can exist in two states: solutions and/or precipitated compounds. The solid solubilities in the aluminum of all alloying elements are limited. When the content of an alloying element exceeds its solid solubility limit, the excess will exist in the form of precipitated dispersions or secondary phases. Alloying elements cause lattice distortion in a solid solution and introduce new interfaces in the precipitated state. For instance, Al-Si alloys contain solid solution Si and eutectic Si phases. The morphology of eutectic Si may be lamellar, acicular, or fibrous [22]. Therefore, the thermal conductivity of aluminum alloys depends on alloying elements and their existing states.

Temperature is another critical factor influencing the thermal conductivity of aluminum alloys. During heat transfer of aluminum alloys, carriers consist of predominant electrons and phonons, and there are barriers of electron–phonon, electron–impurity, phonon–electron, phonon–phonon, and phonon–impurity scatterings [23]. Temperature affects scattering levels and thus the thermal conductivity of aluminum alloys [24].

The theory of thermal conduction of metals can investigate the effect of alloying elements and temperature on the thermal conductivity of aluminum alloys. Moreover, aluminum alloys can be regarded as composites composed of an aluminum matrix and secondary phases. Thus, the effective medium theory (EMT) can be utilized to analyze the effect of the characteristic and morphology of secondary phases on the thermal conductivity of aluminum alloys [25,26]. Compared to experiments, theoretical research is efficient, low-cost, and systematic [27,28]. It is beneficial to the composition and structure design of aluminum alloys with high thermal conductivity.

Industrially, casting and heat treatment processes are important ways to regulate the thermal conductivity and mechanical properties of aluminum alloys [29,30,31]. Chen et al. [29] reported that gravity casting Al-Si-Cu-Fe-Zn alloys have higher thermal conductivity than die castings. Heat treatments include solution treatment, aging treatment, and annealing treatment. Lumley et al. [30,32,33] demonstrated that aged Al-Si alloys have finer and more spherical eutectic Si particles and higher thermal conductivity than as-cast alloys. After annealing, the thermal conductivity of aluminum alloys increases significantly, which is associated with a noticeable decrease in mechanical strength [34,35,36,37]. Additionally, additive manufacturing (AM) is an advanced and essential process in fabricating aluminum alloys, whose parameters play an important role in the thermal conductivity of aluminum alloys. These processes affect the thermal conductivity of aluminum alloys by varying existing states of alloying elements and the morphology of secondary phases.

This review formulates theories of thermal conduction of metals and EMT and then analyzes and summarizes the effect of alloying elements, secondary phases, and temperature on the thermal conductivity of aluminum alloys. Based on the two factors of alloying elements and secondary phases, the influence mechanism and law of casting, heat treatment, and AM processes on the thermal conductivity of aluminum alloys are summarized. This review will provide practical references for developing aluminum alloys with high thermal conductivity.

## 2. Theories of Thermal Conductivity of Aluminum Alloys

### 2.1. Theory of Thermal Conduction of Metals

Heat transfer depends on conduction, radiation, and convection [38]. The heat transfer mechanism of metals is mainly thermal conduction, whose carries include predominant electrons and phonons [39]. Thermal conductivity is a parameter that measures the thermal property of materials and is usually denoted as k in W m^−1^ K^−1^. The thermal conductivity of metals is composed of electronic thermal conductivity $k_{e}$ and phononic thermal conductivity $k_{p}$ [23].
(1)$$k=k_{e}+k_{p}$$

Electrons dominate heat and electricity transfer of metals, and electronic thermal conductivity and electrical conductivity are shown in Equations (2) and (3) [24].
(2)$$k_{e}=\frac{1}{3}C_{v}^{e}v_{F}^{2}\tau_{E}$$
(3)$$\sigma =\frac{ne^{2}}{m}\tau_{E}$$
where $\tau_{E}$ is electronic energy relaxation time, representing the average time an electron loses excess energy. $C_{v}^{e}$ is the contribution of electrons to the specific heat per unit volume. $C_{v}^{e}=\frac{3}{2}nk_{B}$ and $k_{B}$ are the Boltzmann constants. $v_{F}$ is electron group velocity and $\tau_{E}v_{F}$ is mean free path. σ is electrical conductivity, reciprocal of the resistivity ρ. e, m, and n denote the electronic charge, mass, and density, respectively.

The Wiedemann–Franz law proposed the relationship between electronic thermal conductivity, electrical conductivity, and temperature [23].
(4)$$\frac{k_{e}}{\sigma }=L_{0}T$$
where T is the absolute temperature and $L_{0}$ is the Lorentz constant.

According to Equations (1) and (4), the thermal conductivity of aluminum alloys is formulated as Equation (5). Hatch et al. [4] proposed that the Lorentz constant of aluminum alloys is 2.1 × 10^−8^ W Ω K^−2^, and c is 10.5~12.6 W m^−1^ K^−1^.
(5)$$k=\sigma L_{0}T+c=\frac{L_{0}T}{\rho }+c$$

The resistivity of aluminum alloys consists of the intrinsic resistivity ρ(T) and residual resistivity $\rho_{0}$, ρ(T) depends on temperature, and $\rho_{0}$ is related to solid solution alloying elements, precipitates, grain boundaries, dislocations, and vacancies [23].
(6)$$\rho =\rho \left(T\right)+\sum_{i}C_{s}^{i}∆\rho_{s}^{i}+\sum_{i}C_{p}^{i}∆\rho_{p}^{i}+\rho_{g}+\rho_{d}+\rho_{v}$$
where $C_{s}^{i}$ and $C_{p}^{i}$ are the content of the *i*th solid solution and precipitated alloying element, respectively. $∆\rho_{s}^{i}$ and $∆\rho_{p}^{i}$ are resistivity increments caused by 1% of the *i*th solid solution and precipitated alloying element, respectively. $\rho_{g}$, $\rho_{d}$, and $\rho_{v}$ are resistivities of grain boundaries, dislocations, and vacancies, which are much smaller than other resistivities [40,41,42,43].

Many researchers have measured the resistivity increment of pure aluminum generated by 1% alloying elements in a solid solution, as shown in Table 2 [1,44,45,46]. Alloying elements increase the resistivity of pure aluminum to varying degrees, attributed to differences in the outer electronic structure and atom radii between aluminum and alloying elements [4,47]. According to the Wiedemann–Franz law [23], alloying elements reduce the thermal conductivity of pure aluminum to varying degrees. Van horn et al. [44] investigated the effect of alloying elements in a solid solution and precipitated state on the resistivity of pure aluminum. Therefore, after determining the species and content of alloying elements, Equations (5) and (6) are available to predict thermal conductivity and investigate the effect of temperature on the thermal conductivity of aluminum alloys.

The theory of the thermal conduction of metals can be applied to study the effect of alloying elements and temperature on thermal conductivity and to design the composition of aluminum alloys based on the thermal conductivity requirement. However, cast aluminum alloys contain multiple species of alloying elements whose mutual interaction is challenging to quantify. Abundant alloying elements form secondary phases in aluminum alloys, the effect of which the theory of thermal conduction of metals cannot analyze.

### 2.2. Effective Medium Theory for Thermal Conductivity of Aluminum Alloys

Aluminum alloys contain various secondary phases, such as eutectic Si and Al_2_Cu [1]. Commonly used 380 and 319 alloys mainly have eutectic Si and Al_2_Cu phases with volume fractions of about 5~8% and 5%, respectively. So, aluminum alloys can be regarded as composites composed of aluminum matrix and secondary phases. EMT can investigate the thermal conductivity of composites. Therefore, EMT can be applied to analyze the effect of the type, volume fraction, and morphology of secondary phases on the thermal conductivity of aluminum alloys.

Series, parallel, and Maxwell–Eucken models are the most fundamental of the two-phase composite theoretical models [25]. Series and parallel models correspond to two-phase arrangements perpendicular and parallel to the heat flow direction. The two models determine the lower and upper boundaries of effective thermal conductivity, formulated as Equations (7) and (8), respectively. Thus, the effective thermal conductivity of two-phase composites relates to the fraction of the two phases in series and parallel.
(7)$$k_{s}={\left(\frac{v_{1}}{k_{1}}+\frac{v_{2}}{k_{2}}\right)}^{-1}$$
(8)$$k_{p}=v_{1}k_{1}+v_{2}k_{2}$$
where $k_{s}$ and $k_{p}$ represent effective thermal conductivities of series and parallel models and $v_{1}$, $v_{2}$, $k_{1}$, and $k_{2}$ represent volume fractions and thermal conductivities of two phases, respectively.

The eutectic Si phase in Al-Si alloys is mainly lamellar [48]. Helsing et al. [49] proposed that the effective thermal conductivity of lamellar eutectic relates to series and parallel models, as formulated in Equation (9). Based on Helsing’s model, Chen et al. [50] calculated the thermal conductivity of eutectic Al-Si alloys to be 161.5 W m^−1^ K^−1^.
(9)$$k=\frac{1}{4}\left[k_{p}+\sqrt{{k_{p}}^{2}+8k_{p}k_{s}}\right]$$

The Maxwell–Eucken model assumes that spherical particles are isolated and distributed throughout the continuous matrix, and the thermal conductivity is formulated in Equation (10). Hamilton [51] introduced an empirical shape factor into the Maxwell–Eucken model to investigate the effect of discontinuous phase morphology on the thermal conductivity of composites, as shown in Equation (11).
(10)$$k=k_{m}\left[\frac{k_{d}+2k_{m}-2V_{d}\left(k_{m}-k_{d}\right)}{k_{d}+2k_{m}+V_{d}\left(k_{m}-k_{d}\right)}\right]$$
(11)$$k=k_{m}\left[\frac{k_{d}+\left(n-1\right)k_{m}-\left(n-1\right)V_{d}\left(k_{m}-k_{d}\right)}{k_{d}+\left(n-1\right)k_{m}+V_{d}\left(k_{m}-k_{d}\right)}\right]$$
where $k_{m}$, $k_{d}$, $v_{m}$, and $v_{d}$ are thermal conductivities and volume fractions of the matrix and discontinuous phase, respectively. n=3/φ and φ are the sphericity of discontinuous phases.

Series and parallel models define a wide range of thermal conductivity for two-phase composites. Hashin and Shtrikman [52,53] proposed a modified model for the theoretical thermal conductivity limit of two-phase composites, referred to as the H-S model. Supposing $k_{2}>k_{1}$, the upper and lower thermal conductivity of composites are formulated in Equations (12) and (13) according to the H-S model. The upper boundary corresponds to the dispersion of the low-conductive phase in the high-conductive phase, and the lower boundary is the opposite.
(12)$$k_{up}=k_{2}+\frac{1-v_{2}}{\frac{1}{k_{1}-k_{2}}+\frac{v_{2}}{3k_{2}}}$$
(13)$$k_{lo}=k_{1}+\frac{v_{2}}{\frac{1}{k_{2}-k_{1}}+\frac{1-v_{2}}{3k_{1}}}$$

In the literature, several studies have investigated the effect of Si content on the thermal conductivity of Al-Si alloys using the Maxwell–Eucken model [50,54]. Furthermore, Stadler et al. [55] employed the H-S model to analyze this effect. However, calculated values of the Maxwell–Eucken and H-S models deviate from experiments greatly, and neither model considers the morphology of eutectic Si.

EMT can be used to study the effect of the characteristic and morphology of secondary phases on thermal conductivity, facilitating the structure design of aluminum alloys. Nevertheless, it is difficult to determine the thermal conductivity with complex morphology secondary phases in aluminum alloys. For example, the morphology of eutectic Si in Al-Si alloys can transform from lamellar to fibrous by melt modification treatment.

The unit cell model can explore the thermal conductivity of composites with complex structural features through the thermal resistance network method [56]. Based on the unit cell model, Wang et al. [57] calculated the effective thermal conductivity of gray cast iron with a locally interconnected graphite structure. Additionally, many researchers investigated the effective thermal conductivity of composites with cylindrical particles [58], reinforced particles [59], and embedded H-shaped fractal-like tree networks [60]. Accordingly, the unit cell model is very appropriate for exploring the effect of various complex morphologies of secondary phases on the thermal conductivity of aluminum alloys.

## 3. Factors Affecting the Thermal Conductivity of Aluminum Alloys

### 3.1. Alloying Elements

Whether alloying elements in aluminum alloys dissolve in the matrix or exist in the precipitated state, they will hinder the movement of dislocations, enhancing mechanical properties and simultaneously scattering electrons, reducing thermal conductivity [12]. The effect of alloying elements on the thermal conductivity of aluminum alloys relates to their species, existing states, and mutual interaction.

#### 3.1.1. Species of Alloying Elements

Common alloying elements in aluminum alloys include Si, Cu, Mg, Zn, Mn, Ti, Cr, V, Zr, Fe, etc. They can be classified as major alloying elements (Si, Cu, Mg, Zn) and trace alloying elements (Mn, Ti, Cr, V, etc.). Table 3 summarizes the maximum solid solubilities of alloying elements in aluminum and the effect of alloying elements in a solid solution and precipitated state on the resistivity of pure aluminum [4,44].

It is evident in Table 3 that the maximum solid solubilities of alloying elements in aluminum are less than 2%, other than Zn, Mg, and Cu. Alloying elements increase the resistivity of aluminum differently. They increase the resistivity in the solid solution more than in the precipitated state. According to Equations (5) and (6), alloying elements weaken the thermal conductivity of aluminum to varying degrees. In a previous study, the authors investigated the effect of alloying elements in a solid solution on the thermal conductivity of aluminum and demonstrated that the weakening order is Cr > V > Mn > Ti > Zr > Si > Mg > Cu > Zn, as shown in Figure 1 [61].

Major alloying elements (Zn, Cu, Mg, and Si) in solid solutions are the least detrimental to the thermal conductivity of aluminum. A total of 1% of Zn, Cu, Mg, and Si in a solid solution decreases the thermal conductivity of aluminum by about 6, 17, 36, and 54 W m^−1^ K^−1^, respectively [61]. Among them, Si is the most common in aluminum alloys and essential in improving fluidity, reducing casting defects, and enhancing mechanical properties [1,62]. Cu is the primary alloying element, which can enhance the strength of aluminum alloys [4]. Al-Si-Cu alloys commonly used for heatproof components contain plenty of Si and Cu [63]. The thermal conductivity of binary Al-Si and Al-Cu alloys decreases significantly with the increasing content of alloying elements [8,50,64].

It can be found in Table 2 and Table 3 that Zn causes the minimum increment in resistivity of aluminum, i.e., Zn weakens the thermal conductivity of aluminum to the slightest extent. In a solid solution, Mg increases resistivity more than Zn and Cu, and less than Si. Additionally, Mg and Zn have large solid solubilities in aluminum and can thereby substantially dissolve in the matrix and enhance strength [65,66]. However, the large content of dissolved Mg and Zn will lead to the worse thermal conductivity of aluminum alloys [4,61].

Trace alloying elements such as Cr, V, Mn, and Ti in a solid solution significantly increase the resistivity of aluminum, demonstrating that trace alloying elements weaken the thermal conductivity of aluminum seriously [4,61]. Ti is usually added to aluminum as Al-Ti or Al-Ti-B alloys to refine grains and enhance strength [67], where Al-5Ti-1B is the most effective [68]. Zhou et al. [69] investigated the effect of trace alloying elements (Mn, Cr, V) on the thermal conductivity of Al-9Si alloys. They found that 0.1% of Mn, Cr, or V decreases the thermal conductivity of approximately 12~19 W m^−1^ K^−1^. It has been reported that boron treatment would make Ti, Cr, V, Zr, and other trace alloying elements precipitate to reduce their content in aluminum alloys [70].

All alloying elements’ additions weaken the thermal conductivity of aluminum. In a solid solution, the weakening order of alloying elements on thermal conductivity is Cr > V > Mn > Ti > Zr > Si > Mg > Cu > Zn, and trace alloying elements Cr, V, Mn, and Ti have the strongest weakening effect. Therefore, fabricating aluminum alloys with high thermal conductivity should strictly restrain the content of trace alloying elements.

#### 3.1.2. Existing States of Alloying Elements

In Table 3, the effect of alloying elements in a solid solution on the resistivity of aluminum is 2~44 times that in the precipitated state. It indicates that solid solution alloying elements weaken the thermal conductivity of aluminum more significantly than those in the precipitated state. This is because alloying elements in a solid solution generate lattice distortion while introducing new interfaces in the precipitated state, which scatter electrons in different ways. Therefore, the thermal conductivity of aluminum alloys also depends on the existing states of alloying elements.

The authors have explored the effect of major alloying elements (Si, Cu, Mg, Zn) in two states on the thermal conductivity of aluminum, as depicted in Figure 2 [61]. We found variations in thermal conductivities of Al-1.65Si, Al-5.67Cu, Al-1.5Mg, and Al-12Zn, which can reach 73, 73, 21, and 36 W m^−1^ K^−1^, respectively, due to the existing state transformation of alloying elements.

Cast Al-Si alloys with Cu and a small amount of Mg are the most commonly used commercial aluminum alloys for heatproof applications [61,71]. At room temperature, the equilibrium solid solubility of Si is approximately 0.05% in aluminum, so Si is mainly present as the second phase in aluminum alloys [4]. With increasing temperature, the solid solubility of Si in aluminum increases, and Si phases dissolve into the matrix, decreasing the thermal conductivity of aluminum alloys significantly [72]. Mulazimoglu et al. [73] reported that the difference in thermal conductivity of an Al-1.6Si alloy is approximately 36 W m^−1^ K^−1^ when Si is either in a solid solution or in the precipitated state. Therefore, the solid solution and precipitation of Si affect the thermal conductivity of aluminum alloys significantly.

Cu usually exists as the Al_2_Cu phase in aluminum alloys [1]. The maximum solid solubility of Cu is approximately 5.67% in aluminum, so the solid solution and precipitation of Cu are visible [4,36]. The dissolution of Al_2_Cu in the matrix enhances strength but decreases the thermal conductivity of Al-Si-Cu alloys [74,75]. Choi et al. [36] investigated the effect of the dissolution of Al_2_Cu on the thermal conductivity of an Al-4.5Cu alloy. They found that the variation in thermal conductivity can reach 45 W m^−1^ K^−1^ due to the dissolution of Al_2_Cu. Therefore, the solid solution and precipitation of Cu have an essential effect on the thermal conductivity of Al-Cu alloys.

In Al-Si-Mg alloys, Mg and Si will combine and form the Mg_2_Si phase, whose maximum solid solubility is approximately 1.4% in the aluminum matrix [72,76]. Choi et al. [77] found that the dissolution and precipitation of Mg_2_Si in the matrix affect the thermal conductivity of Al-Si-Mg alloys.

The solid solution and precipitation transformation of Si, Cu, and Mg affect the thermal conductivity of aluminum alloys. When alloying elements in a solid solution precipitate from the matrix, the thermal conductivity of aluminum alloys will increase, whereas the dissolution of alloying elements will decrease the thermal conductivity. Therefore, when fabricating aluminum alloys with high thermal conductivity, we can reduce the solid solution content of alloying elements to mitigate their weakening effect on thermal conductivity.

#### 3.1.3. Mutual Interaction of Alloying Elements

Multiple alloying elements are usually added to aluminum alloys to meet the mechanical property requirements. Cast Al-Si alloys usually contain Cu, Mg, and trace alloying elements, such as Fe and Mn.

Maximum solid solubilities of major alloying elements such as Si, Cu, and Mg in aluminum are 1.65%, 5.67%, and 14.9%, respectively [4]. However, the maximum solid solubilities of Si and Cu are 1.1% and 4.8% in Al-Si-Cu alloys, respectively [78], and those of Si and Mg are 1.1% and 0.9% in Al-Si-Mg alloys, respectively [79]. The variation indicates that solid solubilities of Si, Cu, and Mg affect each other in aluminum alloys.

In Al-Si-Mg and Al-Mg-Si alloys, Si and Mg will form the Mg_2_Si phase, and the simultaneous presence of Cu will create Al_4_Cu_2_Mg_8_Si_5_ [80]. The Mg/Si ratio of Al-Mg-Si alloys will affect the solid solution content of Mg and Si, as well as the volume fraction and morphology of Mg_2_Si, both of which influence mechanical and thermal properties [81].

Fe is the main impurity element in cast Al-Si alloys, which usually combines with Si to form needle or flaky β-Al_5_FeSi and script-shaped α-Al_15_Fe_3_Si_2_, α-Al_8_Fe_2_Si, α-Al_12_Fe_3_Si_2_ phases [82]. The β-Al_5_FeSi phase potentially generates stress concentration and harms the mechanical properties of aluminum alloys [83]. After adding an appropriate amount of Mn, a needle β-Al_5_FeSi will transform to a script-shaped α-Al_15_(Fe, Mn)_3_Si_2_, thereby reducing the harmful effect [84].

Gan et al. [85] investigated the effect of Fe on the thermal conductivity of Al-Si alloys. They found that with increasing Fe content, the thermal conductivity of pure aluminum decreases monotonously, while that of Al-Si alloys initially increases and then decreases. More Si content corresponds to higher peak values in the thermal conductivity of Al-Si alloys with 0.3% Fe. The relationship is attributed to combining a small amount of Fe with Si to form β-Al_5_FeSi, which reduces the solid solution content of Si in the matrix and thus improves the thermal conductivity of Al-Si alloys.

Coexisting multiple alloying elements in aluminum alloys will affect the solid solubilities of each element and interact to form intermetallic compounds [4]. Variations in the type and morphology of intermetallic compounds affect the thermal conductivity and mechanical properties of aluminum alloys to some extent.

### 3.2. Secondary Phases

Alloying elements above their solid solubilities will form secondary phases, whose characteristics, volume fraction, and morphology affect the thermal conductivity of aluminum alloys.

#### 3.2.1. Thermal Conductivity of Secondary Phases

Cast Al-Si alloys commonly contain the eutectic Si phase formed during solidification [1]. At 300 K, the thermal conductivity of single crystal Si is 145 W m^−1^ K^−1^ [86], and that of the polycrystalline Si is approximately 15~30 W m^−1^ K^−1^ [87,88]. The value of the polycrystalline Si is always adopted when theoretical deduction or computation concerns the thermal conductivity of eutectic Si [50,55].

Stadler et al. [55] investigated the effect of Cu and Ni on the thermal conductivity of Al-Si alloys. They found that the thermal conductivity of Al-Si alloys decreases monotonously with the increasing content of Cu and Ni. The weakening effect of Ni is greater than that of Cu, which relates to secondary phases formed by Cu and Ni.

Cu is usually present as the Al_2_Cu phase in aluminum alloys. At room temperature, the thermal conductivity of Al_2_Cu is approximately 126 W m^−1^ K^−1^ and decreases to about 108 W m^−1^ K^−1^ with increasing temperature to its melting point [89,90]. The solid solubility of Ni in aluminum is minimal, and Ni usually exists in the form of Al_3_Ni, whose thermal conductivity is about 35 W m^−1^ K^−1^ [91,92]. The thermal conductivity of Al_3_Ni is much lower than that of Al_2_Cu. According to EMT [25], the weakening effect of Al_3_Ni on the thermal conductivity of aluminum alloys is more significant than that of Al_2_Cu. Thus, the weakening effect of Ni is more pronounced than Cu.

A large amount of Cu in aluminum alloys may result in microporosity [93]. The adding range of Cu in Al-Si alloys is commonly less than 3%, which forms a low volume fraction of Al_2_Cu [16]. Aluminum alloys also have small amounts of dispersed Mg_2_Si, α/β-AlFeSi, and so on. These dispersed secondary phases are low-content, small-size, and granular, the effect of which on the thermal conductivity of aluminum alloys can be investigated by the Maxwell model, which does not need to consider morphology. Furthermore, small amounts of dispersed secondary phases have less effect, but their dissolution and precipitation affect the thermal conductivity of aluminum alloys significantly [36,75].

#### 3.2.2. Morphology of Secondary Phases

Aluminum alloys may have a high-volume fraction and large size of secondary phases, whose morphology significantly affects thermal conductivity. In common cast Al-Si alloys, the Si content and volume fraction of eutectic Si range from 4.5~13% and 3~12%, respectively [5]. Therefore, the morphology of eutectic Si affects the thermal conductivity of Al-Si alloys.

Several researchers have demonstrated that adding chemical modifiers (Na, Sr) can modify the morphology of eutectic Si and improve the thermal conductivity and mechanical properties of Al-Si alloys [73,94,95]. The authors investigated the effect of Sr content on the eutectic modification level of Al-7Si alloys [85]. We found that the eutectic modification level increases with increasing Sr content, with 56 ppm Sr leading to the fully modified structure. The microstructure of unmodified and modified eutectic Si of Al-7Si alloys is shown in Figure 3. Eutectic Si in unmodified Al-Si alloys is lamellar and/or acicular, which will transform into fibrous after modification treatment [48,96,97,98].

Gan et al. [8] investigated the effect of Sr modification on the thermal conductivity of Al-Si alloys. They revealed that Sr modification could transform eutectic Si from flaky to fibrous, which was beneficial to improve the thermal conductivity of Al-Si alloys. After modification treatment, the increment in thermal conductivity increased with increased Si content, and the increment of Al-9Si alloys can reach 30 W m^−1^ K^−1^. The increment of an A356 alloy (Al-7Si-0.35Mg) is approximately 20 W m^−1^ K^−1^ [99]. These results signify that the morphology of eutectic Si has an essential effect on the thermal conductivity of Al-Si alloys.

Figure 4 shows the hindrance mechanism of lamellar and fibrous eutectic Si to heat transfer electrons. Lamellar eutectic Si obstructs most electrons, while abundant electrons can pass through fibrous eutectic Si gaps. The heat transfer efficiency of electrons can be improved significantly with eutectic Si transforming from lamellar to fibrous after modification treatment. Thus, the thermal conductivity of Al-Si alloys with fibrous eutectic Si is higher.

Additionally, hypereutectic Al-Si alloys contain eutectic Si and primary Si, the morphology of which can also be changed by modification treatment [4]. The primary Si can be obviously refined using P as the modifier [100]. Jia et al. [101] revealed that the variation in morphology and size of primary Si improves the thermal conductivity of hypereutectic Al-Si alloys after modification treatment.

Series and parallel models can be applied to analyze the effect of various morphologies of the Si phase on the thermal conductivity of Al-Si alloys, enabling the strategic structure design of aluminum alloys. Industrially, modifying the eutectic Si morphology is crucial to fabricate Al-Si alloys with high thermal conductivity and mechanical properties.

### 3.3. Temperature

Temperature affects the heat transfer efficiency of electrons and phonons, thereby affecting the thermal conductivity of aluminum alloys [24]. The thermal conductivity of pure aluminum changes slightly with temperature [2,23,102]. When increasing temperature from 250 K, the thermal conductivity of aluminum slightly rises to 240 W m^−1^ K^−1^ at 400 K and steadily declines to 220 W m^−1^ K^−1^ at approximately 800 K.

Within the range of 25~400 ℃, the thermal conductivity of as-cast and annealed 2A12, 2A50, 6066, and ZL107 (Al-Si-Cu) alloys rise with increasing temperature [103]. There is a similar trend for as-cast and aged A319 (Al-Si-Cu) and A356 (Al-Si-Mg) alloys [104,105]. Lumley et al. [30] demonstrated that when the temperature rises from 25 to 150 ℃, the thermal conductivity of an A380 (Al-Si-Cu) alloy increases in as-cast, T6, and T7 conditions. Choi et al. [71] reported increased thermal conductivity of Al-Si-Cu-Mg alloys with increased temperature. However, the thermal conductivity of eutectic Si [106,107,108], Al_2_Cu [109], and Mg_2_Si [110] of Al-Si-Cu, Al-Si-Mg, and Al-Si-Cu-Mg alloys declines with increasing temperature. Therefore, the change rule of the thermal conductivity of aluminum alloys with temperature depends on the matrix significantly.

Choi et al. [36] demonstrated that the thermal conductivity of Al-4.5Cu alloys varies with increasing temperatures from 25 to 300 ℃ in two ways, attributed to the transformation of existing states of Cu in aluminum. With increasing temperature, the thermal conductivity of an Al-4.5Cu alloy decreases with the solid solution of Cu in the matrix and increases with the precipitation of Cu from the matrix. The thermal conductivity of an Al-4.5Cu alloy is the lowest when Cu is entirely in a solid solution after solution treatment and the highest when Cu is almost completely in the precipitated state after annealing treatment. Therefore, the effect of temperature on the thermal conductivity of aluminum alloys relates to the solid solution and precipitation transformation of alloying elements in the matrix.

The effect of temperature on the thermal conductivity of aluminum alloys depends on the matrix and secondary phases, predominantly the former. Temperature affects the electrons’ scattering and existing states of alloying elements in aluminum. However, the influence mechanism and rule of temperature on the thermal conductivity of aluminum alloys are still unclear.

## 4. The Effect of Processes on the Thermal Conductivity of Aluminum Alloys

### 4.1. Casting Process

The casting processes of aluminum alloys can be divided into gravity casting and die casting, and gravity casting includes sand casting, investment casting, permanent mold casting, and so on [5]. Compared to gravity casting, die casting has the advantages of high efficiency and low cost and can mass-produce large-volume, highly integrated, and complex-shaped castings [33].

The cooling rate of aluminum alloys varies under different casting processes. Many researchers have found that the cooling rate of aluminum alloys affects the microstructure and thermal conductivity significantly [111,112]. For example, the cooling rate of Al-Si alloys influences the size and morphology of eutectic Si [113]. The cooling rate in die casting is higher than in gravity casting, and the secondary phases of die casting Al-Si-Cu-Fe-Zn alloys are noticeably finer than those of gravity casting alloys, as shown in Figure 5 [29]. High cooling rates can reduce the porosity in aluminum alloys [13]. Chen et al. [29] reported that the thermal conductivity of aluminum alloys decreases monotonously with increasing porosity. Vandersluis et al. [9] demonstrated that higher cooling rates lead to lower porosity, finer eutectic Si, and higher thermal conductivity of 319 alloys.

Today, over 50% of cast aluminum alloys are fabricated by die casting [5,33]. Higher pressure in die casting leads to lower porosity and higher thermal conductivity of aluminum alloys [114]. Cao et al. [115] found that a higher vacuum in high-pressure die casting contributes to lower porosity and smaller pores in Al-9Si-3Cu alloys. Therefore, adjusting die casting pressure and vacuum can improve the thermal conductivity of aluminum alloys by modifying the microstructure.

### 4.2. Heat Treatment

Heat treatments of aluminum alloys mainly include solution treatment, aging treatment, and annealing treatment. Aging treatment mainly contains T4–T7, whose procedures are shown in Table 4 [4,116]. Heat treatments can improve the microstructure and release stress, thus enhancing the properties of aluminum alloys [33,76].

#### 4.2.1. Solution Treatment

Solution treatment is one of the most essential strengthening processes of aluminum alloys. Solution treatment is the process of heating aluminum alloys to a pre-set high temperature and maintaining them for some time, and then quenching the alloys to achieve uniform over-saturated solids [76].

Vandersluis et al. [117] demonstrated that the Al_2_Cu phase of an Al-6Si-3Cu alloy dissolves noticeably after solution treatment, and the fraction decreases from 2.5% to 0.5%. In Al-Si-Mg alloys, Mg_2_Si dissolves in the aluminum matrix during solution treatment [72,76]. Therefore, solution treatment will reduce the thermal conductivity of aluminum alloys significantly due to the dissolution of secondary phases into the matrix.

During solution treatment, Al-Si-Cu and Al-Si-Mg alloys show not only the dissolution of secondary phases but also the evolution of fracture, spheroidization, and coarsening of eutectic Si [75,76,118]. Li et al. [118] found that after solution treatment, the thermal conductivity of an Al-7Si alloy increases by about 10% compared to an as-cast alloy, with increased sphericity of eutectic Si. The increased sphericity of eutectic Si will reduce the resistance to heat transfer electrons in aluminum alloys.

With increasing solution treatment time, the thermal conductivity of Al-Si-Cu alloys initially decreases and then increases [75]. In contrast, the thermal conductivity of Al-Si-Mg alloys first increases and then falls [72]. The difference is attributed to the synergistic effect of the dissolution of secondary phases and variation in the morphology of eutectic Si. The former will reduce the thermal conductivity, and the latter (increased sphericity of eutectic Si) will increase the thermal conductivity.

Therefore, the effect of solution treatment on the thermal conductivity of aluminum alloys depends on the dissolution and variation in the morphology of secondary phases.

#### 4.2.2. Aging Treatment

Aging treatment is essential for fabricating high-performance aluminum alloys, including natural and artificial aging [76]. Artificial aging employs higher temperatures and shorter times than natural aging [119]. Aging treatment aims to precipitate out over-saturated alloying elements in the aluminum matrix to form fine dispersions [4,120]. These precipitated dispersions significantly hinder the movement of dislocations, so aging treatment can enhance the hardness and strength of aluminum alloys [14].

After aging treatment, the precipitation of alloying elements mitigates the lattice distortion and reduces the resistance to electrons, thereby increasing the thermal conductivity of aluminum alloys [32]. Kim et al. [121] found that after aging at 180 ℃ for 5 h, the thermal conductivity of an Al-6.5Si-0.4Mg alloy is higher than that of the as-quenched alloy due to the precipitation of Si and Mg_2_Si. Lumley et al. [30] demonstrated that the thermal conductivity of an aged A380 alloy is approximately 20% higher than that of the as-cast alloy. Chen et al. [29] reported that after solution treatment at 500 ℃ and aging for 4 h, the thermal conductivity of a die casting Al-10Si-0.6Fe-0.7Zn alloy increases from 126.8 to 151.6 W m^−1^ K^−1^.

The artificial aging temperature of aluminum alloys ranges from 150 to 250 ℃, and the time varies from about 6 to 12 h [116]. With increasing aging time, the hardness and tensile strength of aluminum alloys first rise and then decrease, and peaks depend on the aging temperature [122,123,124]. The change rule is attributed to the temperature affecting precipitation rates of alloying elements. Choi et al. [77] demonstrated that high aging temperature for Al-6Si-0.4Mg alloys contributes to the depletion of Si and Mg in a solid solution. They found that the thermal conductivity of an aged Al-6Si-0.4Mg alloy is higher at an aging temperature of 220 ℃ than that at 180 ℃. Esmaeili et al. [125] demonstrated that aging treatment decreases the resistivity of an AA6111 alloy and higher aging temperature results in a faster reduction. According to the Wiedemann–Franz law [23], the thermal conductivity of aluminum alloys increases with increasing aging time, and a higher aging temperature corresponds to a faster increase.

Aging treatment can precipitate out alloying elements in a solid solution, reducing increments in the resistivity of aluminum. Aging treatment can improve the thermal conductivity of aluminum alloys. Furthermore, appropriate aging temperature and time enable the fabrication of aluminum alloys with high thermal conductivity and good mechanical properties.

#### 4.2.3. Annealing Treatment

Annealing treatment of aluminum alloys causes a noticeable depletion of alloying elements in a solid solution to form stable precipitated dispersions [126].

Lin et al. [34] found that after annealing treatment, the tensile strength and yield strength of 5058 aluminum alloys decreased, and the reduction in tensile strength can reach 200 MPa. Similarly, annealing treatment reduces the hardness and tensile strength of Al-Er-Y [35], AA2219 (Al-Cu-Mn) [127], and Al-Er-Yb-Sc alloys [128]. Al-Mg-Si alloy cables show a decrease in yield strength and a significant increase in thermal conductivity after annealing treatment [129]. The thermal conductivity of annealed Al-Er-Yb-Sc alloys increases, which is related to the reduction in dislocations and precipitation of alloying elements in the matrix [128].

Annealing treatment can improve the thermal conductivity of aluminum alloys. It has been reported that the thermal conductivity of annealed Al-Si, Al-Cu, Al-Si-Cu, and Al-Fe-Co alloys is higher than as-cast and solution-treated alloys [36,37,103,130]. Rauta et al. [126] demonstrated that the thermal conductivity of annealed Al-12Si and Al-9Si-3Cu alloys are 60 and 51 W m^−1^ K^−1^ higher than as-cast alloys, respectively. After annealing treatment, the content of solid solution Si decreases in Al-Si alloys [37,131], and Cu in a solid solution precipitates out to form Al_2_Cu in Al-Cu alloys [36]. After annealing, the precipitation of Si and Cu from a solid solution will reduce the resistance to heat transfer electrons and improve the thermal conductivity of aluminum alloys.

Annealing temperature and time affect the thermal conductivity and mechanical properties of aluminum alloys, and the annealing temperature is more significant [34]. Higher annealing temperature and longer time result in lower hardness and tensile strength of aluminum alloys [34,132]. With increased annealing temperature, alloying elements in a solid solution precipitate out faster and form more dispersions, which significantly improves the thermal conductivity of aluminum alloys [129,132].

Annealing treatment can precipitate out almost all over-saturated alloying elements in the aluminum matrix, which dramatically reduces the resistance to electrons. Thus, it can improve the thermal conductivity of aluminum alloys to the greatest extent. However, annealing treatment impairs the strength of aluminum alloys, so it should be avoided when confronting requirements of high strength and high thermal conductivity.

### 4.3. Additive Manufacturing

Recently, AM has attracted much attention in fabricating aluminum alloys with outstanding properties based on layer-by-layer manufacturing [133,134]. The high thermal conductivity of aluminum benefits AM for aluminum alloy applications [135]. So far, Al-10Si-Mg and Al-12Si alloys have dominated AM-fabricated aluminum alloys [136].

Among AM techniques, selective laser melting (SLM) is prevalent for metals [136]. SLM parameters will affect the thermal conductivity of aluminum alloys. The high cooling rate of SLM leads to the oversaturation of alloying elements in the aluminum matrix, significantly reducing the thermal conductivity of aluminum alloys [137,138]. Additionally, pores generated during SLM decrease the thermal conductivity of aluminum alloys [139]. Kim [136] demonstrated that the thermal conductivity of an Al-10Si-Mg alloy increases with the increased polar angle of the specimen, shorter hatch spacing, and decreased scan speed during SLM.

SLM-fabricated aluminum alloys possess a very fine microstructure and high mechanical strength [140]. The tensile strength of SLM-fabricated Al-12Si and Al-10Si-Mg alloys are 140 and 80 MPa higher than as-cast alloys, respectively [133,140]. Many researchers have attempted to precipitate alloying elements from the matrix and improve the properties of aluminum alloys through heat treatment [140,141]. Ming et al. [142] demonstrated that the thermal conductivity of an SLM-fabricated Al-7Si-Mg alloy increased after aging and annealing treatments. Butler et al. [139] reported that the annealing treatment contributed to an increment in thermal conductivity of 18~11% for SLM-fabricated Al-10Si-Mg alloys. Therefore, combining the SLM process and heat treatments is beneficial for fabricating aluminum alloys with high thermal conductivity and strength.

Although AM can only utilize limited types of aluminum alloys, it is promising in fabricating aluminum alloys that fit industrial requirements.

## 5. Conclusions and Perspectives

Related theories of thermal conduction in aluminum alloys are first formulated, and the effect of alloying elements, secondary phases, and temperature on the thermal conductivity of aluminum alloys is analyzed. Additionally, the effect of casting, heat treatment, and AM processes on the microstructure and thermal conductivity of aluminum alloys are discussed and summarized.

(1) The theory of thermal conduction of metals helps to study the effect of alloying elements and temperature on the thermal conductivity of aluminum alloys. EMT can be applied to investigate the effect of the characteristic and morphology of secondary phases on the thermal conductivity of aluminum alloys. The two theories can facilitate the strategic design of the composition and structure of aluminum alloys with high thermal conductivity.

(2) Alloying elements are the most critical factor affecting the thermal conductivity of aluminum alloys and reduce thermal conductivity to varying degrees. Alloying elements in a solid solution weakens the thermal conductivity more significantly than in the precipitated state. Coexisting multiple alloying elements affect the solid solubility of each element and interact to form intermetallic compounds. The mutual interaction affects the thermal conductivity of aluminum alloys to some extent.

(3) The characteristic and morphology of secondary phases affect the thermal conductivity of aluminum alloys, and secondary phases with higher thermal conductivity have less impact. The variation in morphology affects the thermal conductivity significantly. Experimentally, modification treatment can dramatically improve the thermal conductivity of Al-Si alloys by adjusting the eutectic Si morphology.

(4) Temperature affects the heat transfer efficiency of electrons and phonons in aluminum alloys. The change rule of the thermal conductivity of aluminum alloys with temperature depends on the matrix and secondary phases, predominantly on the former. However, the influence mechanism and rule of temperature on the thermal conductivity of aluminum alloys are still unclear.

(5) Casting, heat treatment, and AM processes can adjust the microstructure and thermal conductivity of aluminum alloys. Die casting may result in a finer microstructure than gravity casting. Solution treatment can lead to the dissolution and variation in the morphology of secondary phases. Aging and annealing treatments can precipitate out over-saturated alloying elements in the aluminum matrix. Furthermore, an annealing treatment improves the thermal conductivity of aluminum alloys more significantly than an aging treatment while impairing mechanical strength more strongly. SLM-fabricated aluminum alloys show a very fine microstructure. Combining SLM and heat treatment is beneficial for fabricating aluminum alloys with excellent thermal conductivity and strength.

As reviewed above, there are three main factors influencing the thermal conductivity of aluminum alloys. The effect of alloying elements in a solid solution and precipitated state on the thermal conductivity of aluminum has been explored quantitatively [61]. However, the research on the effect of temperature and mutual interaction of alloying elements is still insufficient. Furthermore, it is very urgent to investigate the effect of eutectic Si morphology on the thermal conductivity of Al-Si alloys, which transform from lamellar to fibrous after modification treatment. After determining the effect of these factors on thermal conductivity, industries can better fabricate aluminum alloys with both high thermal conductivity and high strength.

## Figures and Tables

**Figure 1 materials-16-02972-f001:**
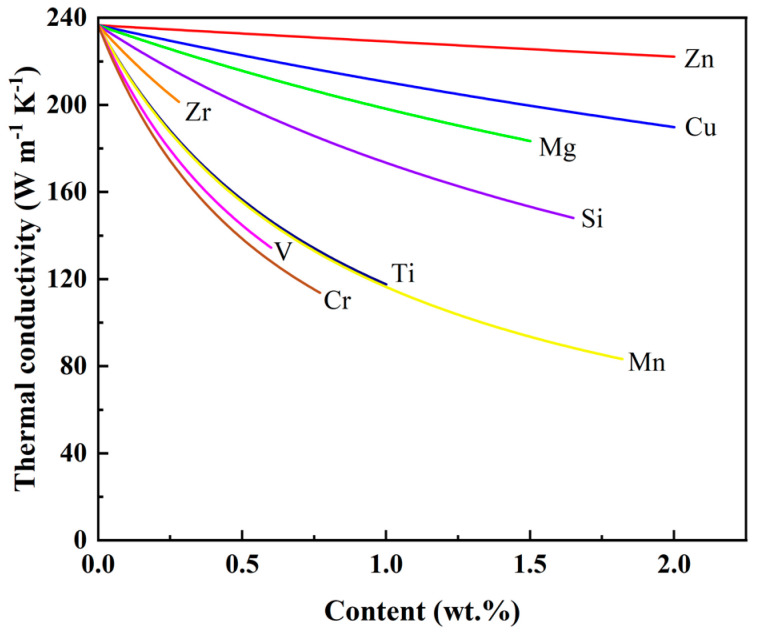
The relationship between the thermal conductivity of aluminum and the content of solid solution alloying elements [61].

**Figure 2 materials-16-02972-f002:**
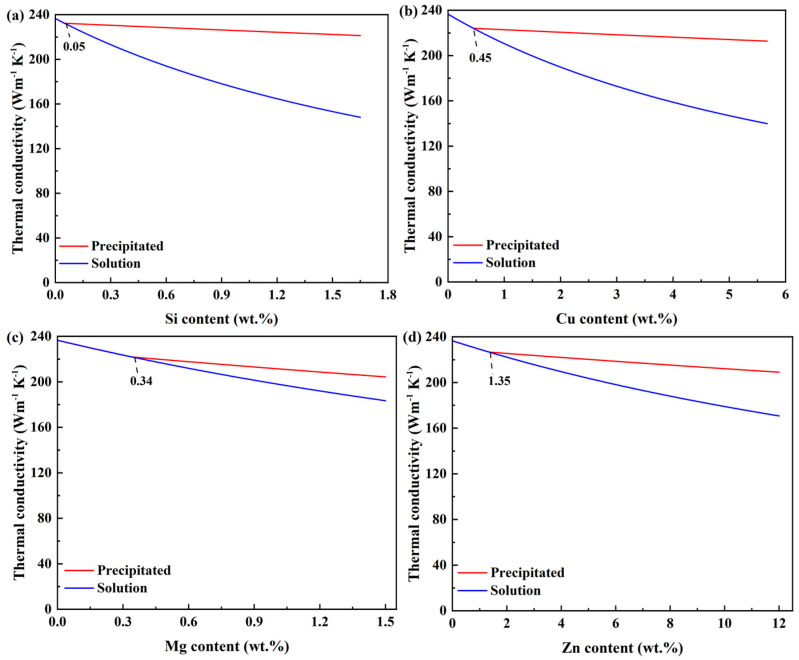
The relationships between the thermal conductivity of aluminum and the content of alloying elements in a solid solution and the precipitated state. (**a**) Al-Si, (**b**) Al-Cu, (**c**) Al-Mg, (**d**) Al-Zn [61].

**Figure 3 materials-16-02972-f003:**
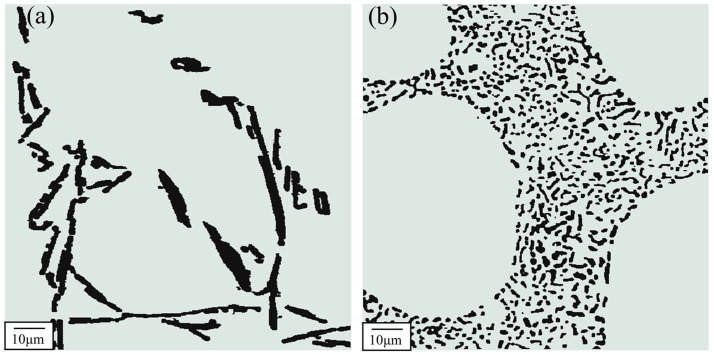
The microstructure of eutectic Si in Al-7Si alloys. (**a**) Unmodified, (**b**) modified [96].

**Figure 4 materials-16-02972-f004:**
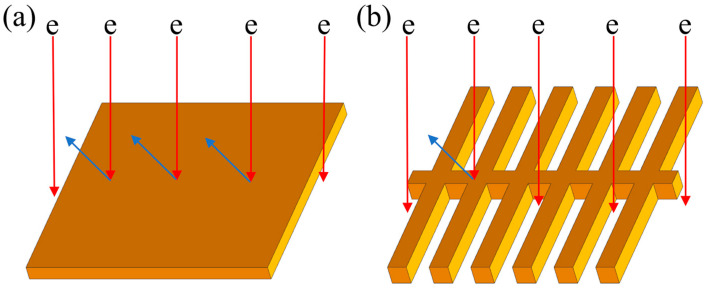
The hindrance of (**a**) lamellar and (**b**) fibrous eutectic Si to electrons.

**Figure 5 materials-16-02972-f005:**
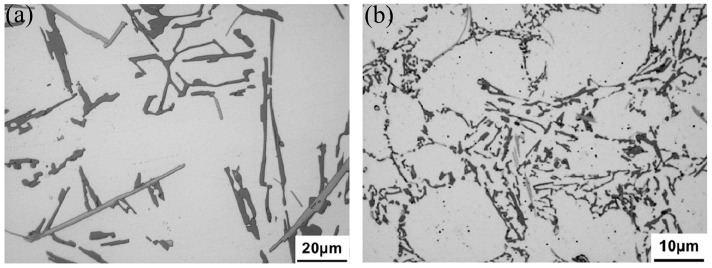
The microstructure of (**a**) gravity casting and (**b**) die casting Al-Si-Cu-Fe-Zn alloys [29].

**Table 1 materials-16-02972-t001:** Thermal conductivity and tensile strength of cast Al-Si alloys at 298 K [2].

Alloys (wt.%)	Temper	Tensile Strength (MPa)	Thermal Conductivity (W m^−1^ K^−1^)
308 (Al-5.5Si-4.5Cu)	F	195	142
319 (Al-6Si-3.5Cu)	T6	280	109
354 (Al-9Si-1.8Cu-0.5Mg)	T6	380	128
355 (Al-5Si-1.3Cu-0.5Mg)	T6	240	152
356 (Al-7Si-0.3Mg)	T6	230	151
357 (Al-7Si-0.5Mg)	T6	262	152
359 (Al-9Si-0.6Mg)	T6	276	138
360 (Al-9.5Si-0.5Mg)	As-cast	305	113
380 (Al-8.5Si-3.5Cu)	As-cast	330	96
383 (Al-10.5Si-2.5Cu)	As-cast	310	96
384 (Al-11.2Si-3.8Cu)	As-cast	330	92
390 (Al-17Si-4.5Cu-0.6Mg)	T7	250	134
413 (Al-12Si)	As-cast	300	121

**Table 2 materials-16-02972-t002:** Increments in resistivity of pure aluminum per 1% alloying elements in a solid solution (μΩ cm/wt.%).

References	Si	Cu	Mg	Mn	Fe	Zn	Cr	Ti	V	Ni	Zr
Van Horn [44]	1.02	0.344	0.54	2.94	2.56	0.09	4.00	2.88	3.58	0.81	1.74
Sacharow [45]		0.4	0.51	2.6	0.41	0.15	3.65		4.56	0.38	1.58
Gauthier [45]	0.47	0.31	0.63	3.8	0.14	0.01	4.7	3.75	3.94	0.09	
CRC-handbook [46]	0.67	0.32	0.5	3.2		0.9	4.42	3.14	4.16	0.05	1.35
Bohner [46]	2	0.4	0.5	3	0.2	0.01	4	/	4	0.2	/
Gauthier [46]	0.37	0.22	0.4	3.3	0.26	0.09	3.6	2.8	4	0.04	/
Willey [1]	1	0.34	0.5	2.9	2.6	0.1	4	2.9	3.6	0.8	1.7
Harrington R.H. [1]	/	0.5	0.6	2.5	0.1	/	3.8	1.8	/	0.1	0.5

**Table 3 materials-16-02972-t003:** Solid solubilities of alloying elements in aluminum and the effect of alloying elements on the resistivity of pure aluminum [4,44].

Elements	Maximum Solubility		Increase in Resistivity (μΩ cm/wt.%)	
	T/℃	wt.%	$\mathrm{Solid}\;\mathrm{Solution}\;∆\rho_{s}^{i}$	$\mathrm{Precipitated}\;∆\rho_{p}^{i}$
Si	577	1.65	1.02	0.088
Cu	548	5.67	0.344	0.03
Mg	451	14.9	0.54	0.22
Zn	382	82.8	0.09	0.023
Mn	660	1.82	2.94	0.34
Ti	665	1.15	2.88	0.12
Cr	660	0.77	4.00	0.18
V	662	0.37	3.58	0.28
Zr	661	0.28	1.74	0.044
Fe	655	0.052	2.56	0.058
Ni	640	0.05	0.81	0.061

The data of Mg is limited to 10% of the maximum solid solubility, and Zn is about 20%.

**Table 4 materials-16-02972-t004:** Commonly used aging treatment for aluminum alloys [4,116].

Designation	Process Procedure
T4	Solution treatment, natural aging
T5	Solution treatment, artificial aging at a low temperature or for a short time
T6	Solution treatment, artificial aging
T7	Solution treatment, overaging/stabilizing

## Data Availability

The data are available in the main text.

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
