# Peer review of "Thermal Conductivity of Aluminum Alloys—A Review"

_materials, 2023, doi:10.3390/ma16082972_

Round 1

Reviewer 1 Report

Dear Authors,

Your Manuscript “thermal conductivity of Aluminum alloys- A Review” presents different data about thermal and electric conductivity of Aluminum alloys but unfortunately you present only one model “the effective medium theory” whereas since your paper is a review you should broaden your interest to other theories such percolation as well as other structural features. In addition to the former main issue there is some style and grammatical mistakes, apart from typos which has to be corrected before resubmission …. some of them are indicated below:

 1)      Please delete the back yellow coloring of certain text in your manuscript before resubmission.

2)       In the abstract, line 15 better start simply your sentence by “Temperature affects also ….”, So please delete “And” which should not be at the starting of sentences.

3)       The authors mentioned only the supporting point of view about the effective medium theory (EMT) and noted that this model could not take into account many morphologies or structural feature without presenting any alternative.

4)       The last part (section 4) lacks figures and representations: it could be helpful to present from literature morphological studies correlated to the change of properties after these different thermal or chemical processes .

Best Regards

Author Response

Dear Reviewer,

Thank you for your comments! We take these comments seriously and have tried to respond to and amend them. All revisions are marked using the “Track Changes” function in the revised manuscript. Our point-by-point responses to the comments are presented in the attachment. 

Thank you again for taking the time to read the manuscript and for your valuable comments, which greatly help improve the quality of the article.

Best wishes

Reviewer 2 Report

This paper comprehensively reviews the thermal conductivity of aluminum alloys. In fact, this article further promotes the industrial design and development of aluminum alloys with high thermal conductivity. The collected information seems good and can attract numerous readers. However, some suggestions and revisions to improve the quality of the manuscript should be applied before acceptance:

1) The manuscript should be revised based on the English language point of view.

2) In the abstract, add the most significant results. It seems that the abstract is not well presented.

3) Some papers should be added to the article and explained such as:

10.1016/j.mtcomm.2022.104301

10.3390/met12081288

10.1016/j.pmatsci.2021.100868

10.3390/met12101627

10.3390/coatings12101559

10.1016/j.msea.2018.06.080

4) It is better to add a section entitled "Future remarks".

5) The effect of main alloying elements on the thermal conductivity of aluminum alloys should be described.

Author Response

(The authors gave the same response as above.)

Reviewer 3 Report

The review paper highlights most important factors, which affect the thermal conductivity of the Al alloy. It is highly recommended to provide more plots, images to support the results, which will be more convincing and interesting for the readers, if it has no restriction for the publication, otherwise it is just a collection of earlier published results. As it is written as a review paper, surprisingly it contains only four images. My comments are given below.

1.     The authors have discussed about the thermal conductivity mostly in as-cast Al-Si alloy. They should also highlight the variation of the thermal conductivity based on the hypoeutectic, eutectic and hyper-eutectic Al-Si alloys as well.

2.     The writing skill should be improved significantly in the revised version.

3.     Line no. 229-230 needs a proper reference.

4.     In many places, the authors used the word, “differently”, which usually makes the sentence very weak. Try to use more scientific terms.  

5.     Line no. 241-242 and 249-250, and 503-504 and 523-524, same sentence has been repeated.

6.     What does it mean by “existing states of alloying elements”. Explain.

7.     It is highly recommended to provide the microstructure of all the systems, that discussed, such as Al-Cu, Al-Mg, Al-Ni, etc., showing the secondary phases, precipitates.

8.     In Line no. 308-309, the authors should show the micrograph of the transformation from needle shaped to script-shaped transformation.

9.      For the section 3.3, the authors need to show some plots of the variation of thermal conductivity with the temperature.

10.   Again, for section 4.1, the authors need to show some kind of plots demonstrating the variation in thermal conductivity of the Al-alloys processed by various methods.

11.   Line no. 528 is not required. In many places, the authors have also discussed about the mechanical properties of the Al-alloys at annealing, which is not necessary to include in this manuscript, as the main theme is based on only thermal conductivity of the Al-alloys, not mechanical properties.

12.   The authors should explain more showing some schematic diagram regarding the change in morphology (flake, fibers or sphericity) on the thermal conductivity.

Regarding the processing methods, now a days, more advanced techniques, such as various additive manufacturing, laser surface remelting of the as-cast alloys have also been using to refine the grain size by varying the cooling rate. The authors need to discuss the thermal conductivity of the Al-alloys processed by these advanced methods, as well.The review paper highlights most important factors, which affect the thermal conductivity of the Al alloy. It is highly recommended to provide more plots, images to support the results, which will be more convincing and interesting for the readers, if it has no restriction for the publication, otherwise it is just a collection of earlier published results. As it is written as a review paper, surprisingly it contains only four images. My comments are given below.

1.     The authors have discussed about the thermal conductivity mostly in as-cast Al-Si alloy. They should also highlight the variation of the thermal conductivity based on the hypoeutectic, eutectic and hyper-eutectic Al-Si alloys as well.

2.     The writing skill should be improved significantly in the revised version.

3.     Line no. 229-230 needs a proper reference.

4.     In many places, the authors used the word, “differently”, which usually makes the sentence very weak. Try to use more scientific terms.  

5.     Line no. 241-242 and 249-250, and 503-504 and 523-524, same sentence has been repeated.

6.     What does it mean by “existing states of alloying elements”. Explain.

7.     It is highly recommended to provide the microstructure of all the systems, that discussed, such as Al-Cu, Al-Mg, Al-Ni, etc., showing the secondary phases, precipitates.

8.     In Line no. 308-309, the authors should show the micrograph of the transformation from needle shaped to script-shaped transformation.

9.      For the section 3.3, the authors need to show some plots of the variation of thermal conductivity with the temperature.

10.   Again, for section 4.1, the authors need to show some kind of plots demonstrating the variation in thermal conductivity of the Al-alloys processed by various methods.

11.   Line no. 528 is not required. In many places, the authors have also discussed about the mechanical properties of the Al-alloys at annealing, which is not necessary to include in this manuscript, as the main theme is based on only thermal conductivity of the Al-alloys, not mechanical properties.

12.   The authors should explain more showing some schematic diagram regarding the change in morphology (flake, fibers or sphericity) on the thermal conductivity.

13. Regarding the processing methods, now a days, more advanced techniques, such as various additive manufacturing, laser surface remelting of the as-cast alloys have also been using to refine the grain size by varying the cooling rate. The authors need to discuss the thermal conductivity of the Al-alloys processed by these advanced methods, as well.

Author Response

(The authors gave the same response as above.)

Round 2

Reviewer 1 Report

The authors corrected their manuscript according to previous recommendations and I congratulate them for this effort. However, there is unfortunately some minor revisions needed before final acceptance for publication which are indicated as following:

1) The authors claim presenting two theories in their review, but the theory about thermal conductivity is based on kinetic theory and that for the electric conductivity is Drude model (free electrons) which are proposed much more than a century ago! Consequently, I suggest that the authors be more specific in the title as following “Effective medium theory for thermal Conductivity of Aluminum Alloys – a review)

2) Please indicate that the heat capacity used in equation 2 is in fact n ce so that one will not have a remaining n in the Wiedemann-Franz law. You can also indicate the expression of well-known electron heat capacity.

3) Please correct the expression of Drude electric conductivity (equation3 in the manuscript) it is proportional to e^2 and not e^3

4) Please replace conclusion and future remarks by conclusion and perspectives

Best Regards

Author Response

Dear Reviewer,

Thank you very much for the detailed corrections! We take these comments seriously and have amend them. All revisions are marked up using the “Track Changes” function in the revised manuscript. Our point-by-point responses to the comments are presented in the attachment.

Thank you again for taking the time to read the manuscript! Your valuable comments and detailed corrections greatly improve the article.

Best wishes to you for good health and pleasant work!

Yours sincerely

Reviewer 3 Report

The authors have extensively revised the manuscript. Therefore, I strongly recommend for acceptance of the manuscript in this reputed journal.

Author Response

Dear Reviewer,

Thank you very much for taking the time to read our manuscript! Your valuable comments and detailed corrections help greatly improve the article.

Best wishes to you for good health and pleasant work!

Yours sincerely